# Greater Tuberosity Fractures after RTSA: A Matched Group Analysis

**DOI:** 10.3390/jcm12031153

**Published:** 2023-02-01

**Authors:** Farah Selman, Philipp Kriechling, Lukas Ernstbrunner, Karl Wieser, Paul Borbas

**Affiliations:** 1Department of Orthopaedic Surgery, Balgrist University Hospital, University of Zurich, 8008 Zurich, Switzerland; 2Department of Orthopaedic Surgery, Royal Melbourne Hospital, Parkville, VIC 3050, Australia; 3Department of Biomedical Engineering, University of Melbourne, Parkville, VIC 3010, Australia

**Keywords:** reverse total shoulder arthroplasty, arthroplasty registry, complication, greater tuberosity fracture, clinical outcome

## Abstract

Periprosthetic fractures, such as acromial and spine fractures, are known complications following implantation of reverse shoulder arthroplasty (RTSA). The entity of greater tuberosity fractures (GTF) has rarely been studied in the literature. The purpose of this study was to analyze the outcome of postoperative greater tuberosity fractures after RTSA compared to a matched control group. The main findings of this study are that a GTF after RTSA is associated with worse clinical outcome scores (mean absolute CS 50 ± 19 (*p* = 0.032); SSV 63% ± 26 (*p* = 0.022); mean force 1 kg ± 2 kg (*p* = 0.044)) compared with the control group (mean absolute CS 62 ± 21; SSV 77% ± 29; mean force 2 kg ± 2 kg). In terms of postoperative range of motion, the fracture group was significantly worse in terms of external rotation (17° ± 19° vs. 30° ± 19° (*p* = 0.029)). Internal rotation, flexion, as well as abduction of the shoulder appear to be unaffected (internal rotation GTF 4 ± 2, control group 5 ± 3 (*p* = 0.138); flexion GTF 102° ± 28°, control group 114° ± 27° (*p* = 0.160); abduction GTF 109° ± 42°, control group 120° ± 39° (*p* = 0.317)).

## 1. Introduction

Since the first case series published by Grammont et al. [1] in 1987, there has been a consistently increasing use of reverse total shoulder arthroplasties (RTSA) worldwide. Due to good clinical outcomes for a variety of indications, such as massive rotator cuff tear, osteoarthritis, fractures, tumors, and for revision surgery [2,3,4,5], RTSA is increasingly replacing the anatomical shoulder prosthesis [6,7]. However, with more frequent use, more complications are being identified. Instability, infections, material loosening, and fractures of the surrounding osseous structures are among the most common complications [7,8,9,10,11]. Isolated periprosthetic fractures, mostly of the acromion, account for approximately 1–5% of all complications and are related to a poorer outcome and higher revision rate [8,12,13,14,15]. The underlying risk factors and pathomechanism of periprosthetic fractures are still incompletely understood. Female sex, osteoporosis, and impaired bone quality in general were reported to have a negative influence [14,16]. Furthermore, previous surgery and increased deltoid length are associated with a higher rate of acromial fractures [14]. The effect of RTSA positioning and design on postoperative fractures has been mainly studied for acromion and scapula fractures and remains controversial [14,17,18,19].

Greater tuberosity fractures (GTF) as a complication after RTSA are rarely mentioned in the literature [13,20]. In fracture arthroplasty for proximal humeral fractures, postoperative union of the greater tuberosity (GT) seems to have an important biomechanical effect. As several studies have shown, correct healing of the GT is associated with better clinical long-term outcomes [21,22,23]. This concerns especially the external rotation function after RTSA [23]. 

Hochreiter et al. [24] recently published that less preoperative to postoperative distalization of the GT is associated with poor functional internal rotation [24]. These findings suggest that the position of the GT relative to the RTSA does influence the postoperative shoulder function. 

Therefore, we hypothesize that RTSA with postoperative GTF is associated with poorer clinical results.

The aim of this study was to compare and report the clinical outcome in patients with and without a fracture of the GT after RTSA.

## 2. Materials and Methods

### 2.1. Patients 

A prospectively enrolled RTSA database of 1360 RTSA implantations between September 2005 and December 2019 was reviewed retrospectively for a GTF. The inclusion criteria were (1) an isolated postoperative greater tuberosity fracture after RTSA, regardless of the indication for RTSA, (2) agreement to be included in the study, (3) age ≥18 years, and (4) a minimum clinical and radiological follow-up period of 2 years after RTSA implantation. The exclusion criteria were (1) a preoperative fracture of the greater tuberosity and (2) further postoperative fractures. The degree of dislocation was specified for each GTF. Displacement was defined as a displacement of the greater tuberosity ≥1 cm, slight displacement if displacement was >3 mm but <1 cm, and no displacement with ≤3 mm displacement on standard radiographs [25]. 

The GTF group was pair matched in a 2:1 manner to a control group with regards to sex, age at surgery, indication for RTSA, length of follow-up, and body mass index.

### 2.2. Surgical Technique

According to the indication, patients were treated with different RTSA designs. The indication for RTSA varied from cuff tear arthropathy, osteoarthritis, proximal humeral fracture, instability, or conversion surgery from failed anatomic prosthesis. Seven of the 17 stems were cemented (41%), the remaining stems were implanted in press-fit technique. All surgeries were performed through a deltopectoral approach by different senior shoulder surgeons as previously described [26]. The subscapularis tendon was detached if needed and sutured/reinserted if possible. 

Postoperatively, all patients had the same standard follow-up treatment.

### 2.3. Clinical and Radiological Evaluation 

All patients were evaluated postoperatively at 6 weeks, 4.5 months, 1 and/or 2 years and then every 2–4 years. At each consultation, functional outcomes were determined with the absolute Constant–Murley score (aCS) and relative Constant–Murley score (rCS), including pain assessment [27], subjective shoulder value (SSV) [28] and range-of-motion (ROM) assessment.

The definitions for minimal clinically important differences (MCIDs) are based on studies by Torrens [29] and Simovitch [30] et al. The MCID cutoff for active flexion is 12°, abduction 7°, and 3° for internal and external rotation. The anchor-based MCID for the Constant score is 5.7 points.

Radiologic follow-up consisted of routine radiographs in 3 standardized planes and, if necessary, additional imaging with computed tomography (CT) or magnetic resonance imaging (MRI). The GTFs were diagnosed by an orthopedic resident, a consultant with a specialization in shoulder surgery, and/or a musculoskeletal radiologist.

### 2.4. Statistical Analysis and Data Collection 

Study data were collected and edited using REDCap (Research Electronic Data Capture) tools provided by Balgrist University Hospital [31,32]. Preoperative and postoperative scores were compared using the Wilcoxon rank sum test. Comparison of 2 groups were performed using the Mann–Whitney-U test. The χ^2^ test was used for categorical variables. A *p*-value of less than 0.05 was considered significant. Owing to the given population of patients with GTFs, no power analysis was carried out. Statistical analysis was performed using SPSS software (version 27.0; IBM, Armonk, NY, USA).

## 3. Results

### 3.1. Incidence and Indication for RTSA 

We identified 19 (1.4%) of 1360 shoulders with a postoperative GTF at a mean of 24 months (±38 months). Two patients (10.5%) had a follow-up <2 years and were excluded. Basic demographic and arthroplasty-related data are summarized in Table 1. 

The GTFs occurred immediately in the first postoperative X-ray (day 0) up to 9 years postoperatively (Figure 1). The day 0 fractures were not detected intraoperatively. Otherwise, they would have been treated during the initial surgery. Further, 16 patients (94%) had no trauma causing the GTF. One patient had a GTF 125 days after RTSA after a fall with unclear trauma mechanism. The atraumatic GTFs were accidentally noticed in a follow-up examination or during work-up of postoperative functional limitations or pain. Thus, we were not able to fully reconstruct the fracture mechanism or exact fracture timing. The fracture time is equated with the radiological diagnosis time. 

Six GTFs (35%) were nondisplaced, five patients (30%) had a slightly displaced GTF, and six (35%) patients had a dislocation of the greater tuberosity.

Most GTFs were treated conservatively (12 out of 17, 70%). Conservative treatment of postoperative GTF included immobilization in a brace for 4–6 weeks with initial passive mobilization and consecutive active mobilization. Five patients (five out of 17, 30%) were treated surgically by means of suture cerclage with FiberWire (Arthrex, Naples, FL, USA) (Figure 2). 

Most GTFs (15 out of 17, 88%) did not further displace, regardless of surgical or conservative treatment. More than half of GTFs (nine out of 17, 53%) showed radiological consolidation at follow-up. One patient’s surgically treated GTF did not heal and was further dislocated. After revision surgery with a second FiberWire refixation, sufficient healing of the GT could be achieved. One patient with severe external rotation malfunction and a GTF nonunion was treated with resection of the GT and a latissimus dorsi tendon transfer. 

Patient data according to GTFs are summarized in Table 2.

### 3.2. Clinical Outcome

Clinical outcome measurements with a minimum follow-up period of two years following RTSA implantation were available for 17 patients (100%) in GTF group and 34 patients (100%) in the control group. Outcome data at two-year follow up were compared in each case.

CSs improved in both groups after RTSA (absolute CS: GTF preoperative 30° ± 17°, postoperative 50° ± 19°, *p* ≤ 0.001; control group preoperative 32° ± 13°, postoperative 62° ± 21°, *p* ≤ 0.001; relative CS: GTF preoperative 36° ± 20°, postoperative 62° ± 21°, *p* ≤ 0.001; control group preoperative 40° ± 15°, postoperative 75° ± 22°, *p* ≤ 0.001). Preoperative CSs were comparable in both groups (absolute CS *p* = 0.508, relative CS *p* = 0.241). However, the control group achieved a significantly higher postoperative CS compared to the GTF group (absolute CS 62° ± 21° vs. 50° ± 19°, *p* = 0.032; relative CS 75° ± 20° vs. 62° ± 21°, *p* = 0.015). 

The same result applies to the SSV, which improved in both groups as a result of the surgical treatment (GTF preoperative 26% ± 16%, postoperative 63% ± 26%, *p* ≤ 0.001; control group preoperative 36% ± 17%, postoperative 77% ± 29%, *p* ≤ 0.001). Postoperative SSV was significantly higher in the control group compared to the GTF group (*p* = 0.022). 

Pain, measured by the CS pain points, was comparable preoperatively (GTF 8 ± 4, control 7 ± 4, *p* = 0.367) and improved in both groups after RTSA (*p* < 0.001) with no significant difference (GTF 12 ± 4, control 14 ± 2, *p* = 0.271).

Mean flexion was comparable preoperatively (GTF 67° ± 42°, control 80° ± 37°, *p* = 0.146) and improved in both groups after RTSA (*p* < 0.001) with no significant difference (GTF 102° ± 28°, control 114° ± 27°, *p* = 0.160).

Analog to flexion, shoulder abduction was comparable preoperatively (GTF 54° ± 31°, control 69° ± 35°, *p* = 0.076) and improved in both groups after RTSA (*p* < 0.001) with no significant difference (GTF 109° ± 42°, control 120° ± 39°, *p* = 0.317).

Mean external rotation was preoperatively comparable in both groups (GTF 32° ± 26°, control 28° ± 21°, *p* = 0.822). Mean external rotation did not significantly change in the control group after RTSA (preoperative 28° ± 21°, postoperative 30° ± 19°, *p* = 0.770), whereas the GTF group had a significant worsening in external rotation after RTSA (preoperative 32° ± 26°, postoperative 17° ± 19°, *p* = 0.010). The postoperative difference in external rotation was significant between both groups (*p* = 0.029).

The internal rotation ability was preoperatively comparable in both groups (GTF 4 ± 2, control 4 ± 3, *p* = 0.965) and could not be positively influenced by the prosthesis in both groups (GTF *p* = 0.627, control *p* = 0.137).

The same results appear for preoperative mean force in both groups (GTF 1 kg ± 1 kg, control 0 kg ± 1 kg, *p* = 0.155). The control group showed significantly more force after RTSA (2 kg ± 2 kg, *p* ≤ 0.001), whereas the GTF group did not improve (1 kg ± 2 kg, *p* = 0.441). The difference in postoperative mean force between both groups was statistically significant (*p* = 0.004). 

The detailed outcome analysis is provided in Table 3.

## 4. Discussion

The most important finding of our study is that patients with a postoperative GTF had worse external rotation after RTSA compared to a matched control group.

The difference in external rotation may have a clinical impact, as the postoperative SSV and Constant scores were significantly higher in the control group compared to the GTF group. Additionally, according to Simovitch et al. [30] a difference of 3° ± 2° in external rotation may be clinically relevant. These findings are not unexpected, as it has been shown previously that healing of the GT is relevant to achieve a clinical outcome and the external rotation ability after fracture RTSA [21,22,23,33,34,35,36,37,38]. Ohl et al. [36] concluded in a multicenter study with 420 patients that anatomic healing of the GT in fracture RTSA improved objective and subjective outcome. Excision of the GT is associated with worse functional outcome and increased risk of postoperative instability [36]. Gallinet et al. [22] compared two cohorts of fracture RTSAs with and without GT-fixation retrospectively. They concluded that anatomic fixation of the GT led to better active range of motion, especially in external rotation with an improvement of +35°. These results show the importance of an intact and correct positioned GT for external rotation, as this can only be provided by functioning infraspinatus and teres minor muscles. 

In our cohort, there was a non-significant difference in postoperative flexion and abduction. It seems that these results are not consistent with other studies showing a significant difference in elevation after resection or malunion of the GT [36,37,38,39]. Gallinet et al. [37] reported a significant mean difference of 14° in forward elevation comparing overall population (115° ± 29°) to a group, where the GT was resected (101° ± 25°). Gunst et al. [39] showed a difference of 19° in a comparable study (*p* = 0.04). Ohl et al. [36] had a mean difference of 27° between anatomical healing of the GT (128° ± 28°) and resection (101° ± 25°). We reported an overall difference in flexion and abduction of approximately 10°. Despite the lack of statistical significance in our comparably small study group with postoperative GTF, we agree that active elevation may be affected negatively by a GTF.

The overall clinical outcomes in this series of RTSA are comparable with reported data from previous studies [38,40,41,42]. The GTF group showed significant pain relief and improvement in all functional outcome parameters, except for active external and internal rotation. As expected, the comparative matching group was also unable to show any improvement in internal rotation after RTSA. These results are in agreement with the data of Hochreiter et al. [24], who pointed out factors influencing frequently observed limited internal rotation after RTSA. 

Most of the GTF patients were treated conservatively, regardless of the grade of dislocation. Further, 50% of conservative treated patients achieved consolidation of the GTF during follow up. It remains unclear whether surgical treatment of GTFs can achieve a significant improvement of GTF healing and external rotation function, especially because the fractures occurred at a wide timeline. It is conceivable that, after a certain time, the muscles responsible for external rotation cannot be sufficiently activated despite refixation of the GT. Successful anatomical healing of the GT after surgical treatment is difficult to achieve. Gallinet et al. [22] and Sebastià-Forceda et al. [43] reported only 64% of anatomic healing utilizing suture cerclage fixation as described by Boileau et al. [44]. Our cohort showed similar results with 60% of GTF healing after surgical treatment. 

The exact incidence of a GTF as a complication after RTSA remains unclear. In our database, an incidence of 1.4% was observed. If GTFs occur unnoticed in the postoperative follow-up or do not cause any complaints, these fractures might frequently remain undetected and are, therefore, potentially underreported. 

The specific cause and risk factors for GTFs were not investigated in this study. However, all fractures except one occurred without a trauma history. Therefore, we hypothesize that these fractures either develop from an intraoperative fissure during the implantation of the stem or due to stress shielding in elderly patient with poor bone quality. Especially female patients are at risk of sustaining a GTF, similar to postoperative fractures of the acromion and scapular spine, with 76% (13 out of 17) females in our series.

The main limitations of the study are the inconsistent patient cohort with GTF and the retrospective study design. Due to different indications, different RTSA glenoid types and revision systems were chosen and implanted by different shoulder surgeons. However, the implanted stem design was the same (ZimmerBiomet Anatomical Inverse/Reverse onlay design) in all patients. We further tried to minimalize the bias by the study design with a 2:1 matching. 

At the follow-up appointments, GTF was assessed on plain radiographs and computed tomography scans were not performed routinely. Non-displaced fractures may have been underreported. 

Despite these limitations, the findings of this study add a valuable aspect to scientific knowledge in RTSA complications. 

## 5. Conclusions

In this series of primary and secondary RTSA, the rare complication of a greater tuberosity fracture is significantly associated with an impaired external rotation, less abduction strength, and inferior clinical outcome scores compared to a matched control group. 

## Figures and Tables

**Figure 1 jcm-12-01153-f001:**
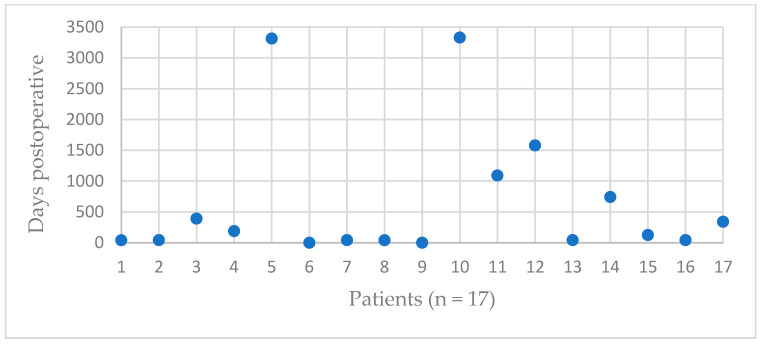
Occurrence of GTP depending on the time.

**Figure 2 jcm-12-01153-f002:**
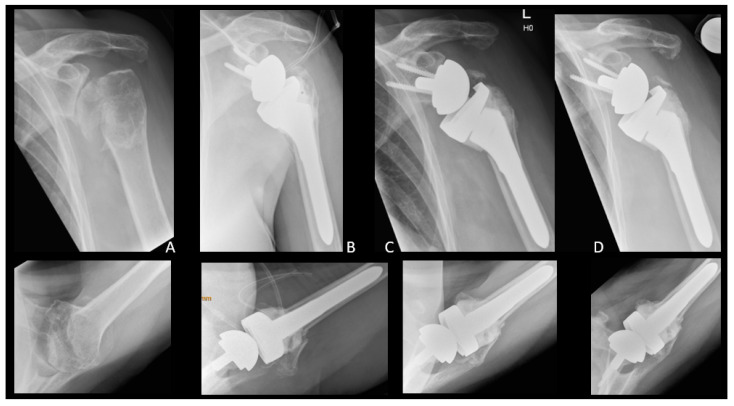
Surgical treatment of a postoperative GTF with FiberWire suture cerclage. Left shoulder, X-rays in a.p. and axial views. (**A**) Preoperative arthritis after humeral head fracture. (**B**) Intraoperative control of RTSA. (**C**) Greater Tuberosity fracture. (**D**) Greater Tuberosity after surgical suture cerclage, unconsolidated, but with no further dislocation.

**Table 1 jcm-12-01153-t001:** Basic demographic and arthroplasty-related data. Mann-Whitney-U testing (*) and Chi square testing (#) were used to compare the groups. Abbreviations: ASA, American Society of Anesthesiologists; CTA, cuff tear arthropathy; MRCT, massive rotator cuff tear; SD, standard deviation.

Characteristic	Fracture	Control	*p*-Value
**Patients, n**	17	34	
**Mean age, years (SD)**	69 (9)	69 (10)	0.984 *
**Male, n (%)**	4 (24)	6 (17)	0.618 #
**Right side, n (%)**	13 (76)	17 (50)	0.070 #
**ASA grade, n (%)**			0.077 *
I	0	1	
II	9	25	
III	8	8	
**Indication**			1.000 #
CTA	1	2	
MRCT without osteoarthritis	1	2	
MRCT with osteoarthritis	1	2	
Osteoarthritis	4	8	
Instability	1	2	
Fracture	3	6	
Arthroplasty revision	6	12	
**Mean BMI, kg/m^2^ (SD)**	29 (7)	28 (6)	0.704 *
**Mean FU, moths (SD)**	72 (37)	84 (35)	0.263 *

**Table 2 jcm-12-01153-t002:** GTF-related data.

Patient	Occurrence of GTF [Days after RTSA]	Trauma	Dislocation	Treatment	Further Dislocation of GTF	Radiological Consolidation of GTF
1	41	No	Slightly	Operative	No	Yes
2	43	No	No	Operative	Yes	Yes, after 2nd refixation
3	391	No	Slightly	Conservative	No	No
4	188	No	No	Conservative	No	No
5	3313	No	Yes	Conservative	No	Partial
6	0	No	Yes	Conservative	No	No
7	43	No	No	Conservative	No	Yes
8	41	No	Yes	Conservative	No	Yes
9	0	No	Yes	Operative	Yes	No
10	3329	No	Slightly	Conservative	No	Yes
11	1091	No	No	Conservative	No	Yes
12	1580	No	No	Operative	No	Yes
13	43	No	Slightly	Operative	No	No
14	743	No	Yes	Conservative	No	No
15	125	Yes	Yes	Conservative	No	No
16	43	No	No	Conservative	No	No
17	340	No	Slightly	Conservative	No	Yes

**Table 3 jcm-12-01153-t003:** Clinical outcome comparing greater tuberosity fracture following RTSA with the control group. Wilcoxon-Ranksum testing ^1^ and Mann-Whitney-U testing ^2^ were used to compare the groups. Abbreviations: CS, Constant Score; ER, External rotation; FUP, Follow-up; IR, internal rotation; kg, kilograms; SD, Standard deviation; SSV, Subjective Shoulder Value.

		Fracture	Control	*p*-Value ^2^
Number		17	34	
Mean CS absolute, points (SD)	Preop	30 (17)	32 (13)	0.508
	Postop	50 (19)	62 (21)	0.032
	*p*-Value ^1^	<0.001	<0.001	
Mean CS relative, % (SD)	Preop	36 (20)	40 (15)	0.241
	Postop	62 (21)	75 (20)	0.015
	*p*-Value ^1^	<0.001	<0.001	
Mean SSV, % (SD)	Preop	26 (16)	36 (17)	0.101
	Postop	63 (26)	77 (29)	0.022
	*p*-Value ^1^	<0.001	<0.001	
Mean CS pain, points (SD)	Preop	8 (4)	7 (4)	0.367
	Postop	12 (4)	14 (2)	0.271
	*p*-Value ^1^	<0.001	<0.001	
Mean flexion, ° (SD)	Preop	67 (42)	80 (37)	0.146
	Postop	102 (28)	114 (27)	0.160
	*p*-Value ^1^	<0.001	<0.001	
Mean abduction, ° (SD)	Preop	54 (31)	69 (35)	0.076
	Postop	109 (42)	120 (39)	0.317
	*p*-Value ^1^	<0.001	<0.001	
Mean ER, ° (SD)	Preop	32 (26)	28 (21)	0.822
	Postop	17 (19)	30 (19)	0.029
	*p*-Value ^1^	0.010	0.770	
Mean IR, ° (SD)	Preop	4 (2)	4 (3)	0.965
	Postop	4 (2)	5 (3)	0.138
	*p*-Value ^1^	0.627	0.137	
Mean force, kg (SD)	Preop	1 (1)	0 (1)	0.155
	Postop	1 (2)	2 (2)	0.044
	*p*-Value ^1^	0.441	<0.001	

## Data Availability

Data will be provided by the corresponding author upon request.

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
