# Peer review of "Greater Tuberosity Fractures after RTSA: A Matched Group Analysis"

_jcm, 2023, doi:10.3390/jcm12031153_

Round 1

Reviewer 1 Report

Thank you for submitting this article. The authors provide data about greater tuberosity fractures after reverse shoulder arthroplasty. There is little data on this topic in the current literature, so the study question is well chosen. There is a clear result shown to have clinical relevance.

The introduction is very informative and covers the current literature and why this topic is worth looking into.

Materials and Methods is well structured. Unfortunately, it is lacking a clear statement about its retrospective character as pointed out in the discussion.

Results: Aside of some suggestions for improvement, this section is fine:

·      Line 105: The question arises if immediately postoperative GTFs (day 0) were maybe intraoperative complications, maybe an explanation can be added.

·      Line 124: Which tendon transfer was performed?

·      Lines 130ff: which follow-up time was compared (always the last available follow-up examination?) 

·      Line 139: SSV is stated as °, it should be %

·      Line 141: Statement about difference of preoperative values between both groups is missing

·      Line 169: Caption and description of figures should be below the image itself

·      There is an inconsistency between figure 1 and table 2: in Figure 1, Patient 6 and 15 are those with GTF at day 0, while in table 2, these are Patients 6 and 9. The numbers of the Patients in these two figures/images generally don’t seem to match.

Discussion: very well written and of adequate length

Conclusion: clear statement supported by the data

Reviewer 2 Report

First of all, I would like to thank for giving me the opportunity to review this manuscript.
This study aimed to analyze the outcome of postoperative greater tuberosity fractures after RTSA compared to a matched control group. The authors have dealt with a rarely investigated “hot topic” which is suitable for the Special Issue and might be interesting for the reader.
The manuscript is well written with appropriate background information and a concise summary of prior literature. The purpose of the study is clearly stated. The results were reported in a clear, concise fashion augmented by tables.

I congratulate the authors for this great work and I have only these minor notes:

Line 75: “subscapularis” instead of “Subscapularis tendon was detached…

Line 174: “…with no without further dislocation.” Please check this sentence

Table 2+3: Please try to remove the red correction markings of Microsoft Word
